# Assessment of Postural Balance in Women Treated for Breast Cancer

**DOI:** 10.3390/medicina56100505

**Published:** 2020-09-27

**Authors:** Iwona Głowacka-Mrotek, Magdalena Tarkowska, Tomasz Nowikiewicz, Magdalena Hagner-Derengowska, Aleksander Goch

**Affiliations:** 1Department of Rehabilitation, Nicolaus Copernicus University in Toruń, Collegium Medicum in Bydgoszcz, 87-100 Toruń, Poland; 2Department of Physiotherapy, Copernicus University in Toruń, Collegium Medicum in Bydgoszcz, 87-100 Toruń, Poland; magdalena.sowa@cm.umk.pl (M.T.); a.goch@cm.umk.pl (A.G.); 3Department of Surgical Oncology, Copernicus University in Toruń, Collegium Medicum in Bydgoszcz, 87-100 Toruń, Poland; tomasz.nowikiewicz@cm.umk.pl; 4Department of Phisical Culture, Nicolaus Copernicus Uniwersity in Toruń, 87-100 Toruń, Poland; m.hagner-derengowska@umk.pl

**Keywords:** balance, mastectomy, adverse consequences, breast cancer

## Abstract

*Background and objectives:* Surgery is the primary and most effective treatment of breast cancer. Unilateral mastectomy disrupts the distribution of muscle tension between the right and the left sides of the body. The aim of the study was to evaluate postural balance in patients treated for breast cancer by mastectomy. *Materials and methods:* A controlled clinical study was conducted on 90 patients who have undergone surgical treatment for breast cancer (mastectomy) 5–6 years prior (Breast Group—BG). The control group (CG) consisted of 74 healthy female volunteers. Analysis of balance was performed using the Alfa stabilography platform. A static test (Romberg’s test) with open and closed eyes was used to assess balance. The following balance parameters were analyzed: path length, statokinesigram area, parameters of deflection and velocity of the foot pressure center. *Results:* The study demonstrated that patients from BG (5–6 years after surgery) obtained worse results in both tests with open (maximum back deviation, maximum forward deviation, average Y deviation, average Y velocity, path length and path surface area) (*p* < 0.05) as well as with closed eyes (maximum backward deviation, maximum forward deviation, mean Y deviation and path length) (*p* < 0.05). *Conclusions:* Our study demonstrated that women 5–6 years after surgery for breast cancer have impaired balance compared to healthy women, despite physiotherapy.

## 1. Introduction

Balance and postural control play an important role in everyday functioning. The sense of balance is responsible for regulating the body’s center of gravity in response to changing surface conditions. Proprioception is responsible for activating postural control mechanisms and their adaptation to new environmental conditions, as well as activating postural mechanisms [1,2]. Breast cancer and the consequences of its surgical and adjuvant treatment may affect the distribution of muscle tone throughout the body. Breast cancer is one of the most common malignancies in women. Surgery remains the primary treatment for this type of cancer. Treatment includes full or partial breast amputation and SLNB (sentinel lymph node biopsy) or ALND (axillary lymph node dissection) surgery. Adjuvant treatment, including chemotherapy, radiotherapy and hormone therapy, are also implemented, depending on the histological type of cancer [3].

The literature describes the following side effects that may occur in women after breast amputation: impaired mobility in the shoulder joint, lymphedema on the operated side, changes in body posture and shape of the feet [4,5,6].

Breast amputation leads to changes in the postural muscle tone and body biomechanics. As a result of treatment, the muscles of the shoulder girdle and upper limbs may weaken. The scar formed after surgery leads to fascial disorders and contractures [7]. Breast amputation is also stressful to the body, which may result in disorders of muscle tone [8]. Pain in the operated breast may also have adverse consequences for maintaining proper body balance [9]. These sequelae may result in balance disorders. Wearing an external breast prosthesis has a positive effect on muscle tone distribution [10]. Studies show, however, that wearing an external prosthesis reduces the quality of women’s everyday functioning and hinders engaging in sports activities [11].

There is little research available in the literature that assesses the effect of unilateral mastectomy on postural balance. The existing studies offer valuable information, but were carried out on a small group of patients shortly after surgery [12]. We have undertaken this study in view of insufficient data on the long-term effects of mastectomy on body balance in women undergoing surgery for breast cancer.

The aim of the study was to assess postural balance in women after 5–6 years following mastectomy due to breast cancer.

## 2. Material and Methods

### 2.1. Study Design

This controlled, non-randomized, clinical study was conducted from February to October 2019 on patients who had undergone unilateral mastectomy 5–6 years prior (Breast Group—BG). Patients that qualified for our study underwent mastectomy between January and December 2014. The study protocol was approved by the Bioethics Committee at the Collegium Medicum in Bydgoszcz (No. KB 20/2019). Mastectomy patients were recruited among women associated with the regional Amazon clubs, who met the inclusion and exclusion criteria.

The control group (CG) consisted of women in a similar age group, meeting the inclusion and exclusion criteria, who voluntarily agreed to participate in the program. The study was conducted in accordance with the Declaration of Helsinki. All the participants were informed in detail about the study protocol and signed an informed consent form. All patients were right-handed.

Criteria for inclusion in the study for patients with BG:-signing consent to participate in the study;-age over 35 years at the time of surgery;-patients declaring regular use of an external breast prosthesis after surgery;-patients declaring participation in physiotherapy;-patients who have undergone surgery for breast cancer 5–6 years before.

Exclusion criteria included the following:-cognitive dysfunction;-patients diagnosed with diseases of the nervous system (Parkinson’s disease, peripheral nerve palsy) or skeletal system (inflammatory diseases, scoliosis >10 Cobb angle), or rheumatic diseases;-patients with disseminated cancer;-patients with lymphoedema of the limb on the operated side;-patients taking medication that affect body balance.

The inclusion criteria for CG were as follows:-consent to participate in the study.

The exclusion criteria were the same as for BG.

The study protocol was as follows:Medical interview (BG patients: type of surgery, type of adjuvant treatment);Measurement of body weight—without footwear, on the scale of 1 kg accuracy;Height measurement.

BMI (Body Mass Index) was calculated for each patient based on the obtained body mass and height measurements.

Posturography was performed on an ALFA stabilography platform by AC International East Co. Before examination, how the test would be performed was thoroughly explained to the patients and a preliminary test was performed. The course of the test was as follows: patient stood shoeless on the platform. Examination was carried out with external breast prostheses. The heels were placed 8 cm apart and feet were at a 10° angle. The device was systematically calibrated during the test. The patient was standing on the stabilography platform, at first with her eyes open for 30 s, and was subsequently asked by the examiner to close her eyes and she stood with her eyes closed for another 30 s. Measurement was carried out continuously for 60 s. During the test, parameters were measured with open (EO) as well as closed eyes (EC). Closing the eyes after 30 s was meant to evaluate the response of the balance control system to the removal of visual cues.

The following posturography parameters were evaluated:-maximum deviation of the center of foot pressure (mm)—this parameter was calculated for left-sided, right-sided, back and forward deviation;-average deviation of the center of foot pressure in the lateral (X) and anteroposterior (Y) directions;-average velocity (mm/s)—mean velocity at which the foot pressure center moves during the test, average velocities in the lateral (X) and antero-posterior (Y) direction were calculated;-path length (mm)—the length of the path traveled by the center of pressure;-surface area (mm^2^)—this parameter is calculated by connecting the points of the statokinesigram with lines.

### 2.2. Statistical Analysis

The analyses were conducted using IBM SPSS Statistics v. 25. In the first step, descriptive statistics were analyzed together with the Shapiro–Wilk distribution test for normal distribution. A *t*-test for independent variables or U Mann–Whitney test were used for comparison of the two groups depending on the distribution of variables, while a Wilcoxon signed-rank test was used for comparison of two variables within one group.

Test probability at *p* < 0.05 was considered significant, and highly significant at *p* < 0.01.

## 3. Results

### 3.1. Group Characteristics

There were 169 women included in the study; 74 in the control group (CG) and 90 in the study group (BG). There were no statistically significant differences between the groups with regard to age, body weight, height or BMI (*p* > 0.005). The study group was also characterized with respect to the type of surgery of the axillary fossa, the operated breast, as well as the type of adjuvant treatment. Detailed clinical and sociodemographic characteristics are presented in Table 1.

### 3.2. Comparison of Open-Eye and Closed-Eye Romberg’s Test Results in Patients from BG

Using the Wilcoxon’s test, we performed an analysis of the results of Romberg’s test with eyes open vs. closed among the study group patients. The results are demonstrated in Table 2.

Analysis showed that BG patients achieved higher values of maximum deviation to the left, right, backward, forward, mean X and Y velocities and path length when the test was performed with closed vs. open eyes (*p* < 0.001). Results did not differ significantly between open-eye vs. closed-eye tests with regard to the mean X and Y deviation or the surface area.

### 3.3. Comparison of Open-Eye vs. Closed-Eye Romberg’s Test Results among Patients from CG

Analogous analyses using Wilcoxon’s test were carried out to compare the results of Romberg’s test with eyes open vs. closed among patients from the CG group. The results of the analysis are shown in Table 3.

In the CG group, patients obtained higher scores for maximal right-sided deviation (*p* = 0.001), average velocity X (*p* < 0.001) and Y (*p* < 0.001), as well as path length in the Romberg’s test with eyes closed vs. eyes open. For the remaining parameters, the differences between the open- and closed-eye samples were not significant (*p* < 0.05).

### 3.4. Comparison of Romberg Test Results in BG and CG Groups

In the next step, the U Mann–Whitney test was used to compare the results of Romberg’s test in the study group vs. control group. The results of the open-eye test are presented in Table 4, while the closed-eye test results are in Table 5.

With open eyes, the value for maximum left-sided deviation was higher in CG than in BG. Patients with BG obtained higher scores for maximum backward deviation (*p* = 0.003), forward deviation (*p* = 0.029), mean Y deviation (*p* = 0.007), average velocity Y (*p* = 0.005), path length (*p* = 0.035) and surface area (*p* = 0.009) compared to patients from the control group. Both groups did not differ with regard to maximum deviation to the right, mean deviation X or mean velocity X (*p* > 0.05).

In the test with the closed eyes, women from the study group obtained higher results for maximum backward deviation (*p* = 0.007), forward deviation (*p* = 0.015), mean Y deviation (*p* = 0.009) and path length (*p* = 0.046) compared to women from the control group. Both groups did not differ with regard to maximum left-sided or right-sided deviation, mean X deviation, X velocity or surface area (*p* > 0.05).

### 3.5. Comparison of Romberg’s Test Results in the MAS Group in Terms of the Operated Breast

We used Mann–Whitney U tests to compare the Romberg’s test results of women from the MAS group who underwent either right or left mastectomy. The results of tests conducted with opened and closed eyes are presented in Table 6.

Detailed analysis of the results showed no significant differences between experimental groups in terms of maximum sways to the left and right in tests performed both with opened and closed eyes. Such results mean that, regardless of the operated breast, women achieve similar outcomes in the analyzed tests.

### 3.6. Comparison of Romberg’s Test Results in the MAS Group in Terms of BMI of Examined Women

We used Mann–Whitney U tests to compare the results of the Romberg’s tests performed with opened and closed eyes in women from the MAS group, whose BMI was either lower or higher than 25. The results of tests performed with opened and closed eyes are presented in Table 7. Detailed analysis of results has shown no significant differences between experimental groups in terms of maximum sways to the left and right in tests performed both with opened and closed eyes. It means that, regardless of BMI, patients from the MAS experimental group achieved similar results in terms of maximum sways to the left and right.

## 4. Discussion

Our study involved the assessment of postural balance in women who had undergone breast amputation for breast cancer. The control group consisted of healthy women of similar ages. The CG and BG groups did not differ with regard to height, weight or BMI.

Statistical analysis showed that women from both the control group and the study group achieved worse balance parameters with eyes closed than with eyes open. These results were in line with those of other authors [12].

In our study, patients did not decide for breast reconstruction. In Poland, only a small number of women decide to undergo breast reconstruction. We explained that IBR (immediate breast reconstruction) is a very rare procedure in Poland, as it only concerns 20–40% of patients. Some reasons why Polish women resign from breast reconstruction include fear of another surgical intervention, fear of surgical complications and hospitalization, age, fear of disease recurrence and financial issues. Women are also inadequately informed about the course of breast reconstruction surgery and its consequences. Some patients do not decide for this procedure due to their advanced age and the fact that breast reconstruction may be perceived as an act of vanity [13].

There are studies available in the literature assessing the effects of breast amputation on body posture, range of joint mobility, occurrence of lymphedema and lung ventilation disorders and changes in kinematics and biomechanics of the upper limb on the operated side as well as the trunk [14,15]. Breast amputation reduces the level of physical activity, and has a negative effect on the musculoskeletal system and coordination [16]. The biomechanics and kinematics of the upper limb and the torso on the operated side change after breast removal. Studies have shown that patients who have undergone a conservative method have had fewer and less severe adverse surgical effects [17]. Changes in body posture occur less frequently and are less advanced in patients who have undergone surgery with simultaneous breast reconstruction. A study by Jeong JH, Choi B at al. showed that 3 months after surgery the scoliosis angle was significantly increased in patients undergoing mastectomy for breast cancer compared to women operated on with simultaneous breast reconstruction [18,19]. The number of bilateral (contralateral prophylactic) mastectomies performed in recent years has risen, especially in the United States [20].

We did not find any studies evaluating the effect of bilateral (contralateral prophylactic) mastectomy on body posture and balance. However, it is reasonable to assume that patients who received this treatment may experience improvement in posture and balance.

The causes of the disturbed statics of the torso in women after mastectomy include surgical scar formation and tissue fibrosis in the operated side, causing changes to the figure; one side may be overloaded due to uneven distribution of body mass. Long-lasting overload can lead to disorders of balance, as shown in our study. Patients from the study group had worse results of body balance tests with both open and closed eyes. Studies by other authors have demonstrated that regular use of an external breast prosthesis as well as sleeping in it reduces adverse effects on body posture, and has a positive effect on body symmetry [21].

Hojan et al. proved in their research that wearing an external breast prosthesis has a positive impact on gait parameters and dorsal extensor muscle activity [22,23]. Other studies have shown that patients undergoing simultaneous breast reconstruction had better body posture [19]. In our study, all patients declared wearing an external breast prosthesis and wore an external prosthesis during examination. Studies by Angin evaluated the effect of lymphedema on balance in women after mastectomy. They have demonstrated that women with lymphedema obtained worse results with regard to postural control [24]. Lymphoedema was an exclusion criterion in our study. Furthermore, patients with BG had a smaller range of left-sided deviation than the CG. This result can be explained by the fact that the majority of patients had right-sided amputation, and it can be assumed that the patient after breast amputation engages the contralateral upper limb more, while sparing the side of the body where amputation had taken place and using the opposite side more. Moreover, if the patient is right-handed and has had left-sided amputation, then the right side is engaged even more, and the left side is neglected. Disorders of balance in women after mastectomy may also be caused by an uneven distribution of the load on the operated side compared to on the side where the breast was preserved [25].

Experimental studies have confirmed that when one side of the body is overloaded, the body’s center of gravity is shifted to the side where the load is greater [26]. In our study, worse results of balance tests were noted in the BG group with both closed and open eyes. Research by Fong et al. has shown that properly selected exercise can improve balance in women after breast amputation [27].

In our study, patients declared participation in physiotherapy, and performing exercises to reduce lymphoedema and improve the range of motion in the joints on the operated side. The standard rehabilitation program in Poland does not take into consideration the implementation of exercises influencing coordination and balance in women after mastectomy.

Despite assessing balance long after surgery, our studies have limitations. Posturography parameters were not assessed in our patient study group before surgery. Therefore, we decided to include a control group.

## 5. Conclusions

Our studies have shown that patients 5–6 years after breast amputation surgery achieved worse parameters of balance, both with open (maximum backward deviation, maximum forward deviation, average Y deviation, average Y velocity, path length and path surface area) as well as closed eyes (maximum backward deviation, maximum forward deviation, mean Y deviation and path length).

## Figures and Tables

**Table 1 medicina-56-00505-t001:** Sociodemographic and clinical characteristics of the group.

Variable	CG (*n* = 74)	BG (*n* = 90)	*t*-test	P *
Age–median (SD)	66.78 (4.78)	65.63 (7.86)	1.10	0.272
Weight–median (SD)	73.96 (19.07)	72.22 (10.15)	−0.47	0.641
Height–median (SD)	161.91 (6.93)	162.37 (6.57)	−0.42	0.669
BMI–median (SD)	27.18 (4.41)	27.44 (3.95)	−0.40	0.686
Type of procedure involving lymph nodes *n* (%)		ALND	75 (83.3)	
		SLNB	15 (16.7)	
Operated side *n* (%)		R	51 (50.0)	
		L	39 (50.0)	
Adjuvant treatment		CHTH,	26	
		HTH	9	
		RTH	25	
		CHTH, RTH	30	

SD—Standard Deviation, BMI—Body Mass Index, CG—Control Group, BG—Breast Group, ALND—Axillary Lymph Node Dissection, SLND—Sentinel Lymph Node Dissection, R—Right, L—Left, CHTH—Chemotherapy, RTH—Radiotherapy; HTH—Hormone Therapy, P—statistical significance levels, * U Mann–Whitney test.

**Table 2 medicina-56-00505-t002:** Results of statistical analysis of the Romberg’s test (eyes open vs. closed) among patients from the study group (BG).

Romberg’s Test	Eyes Open			Eyes Closed			*Z*	*p*	*r*
	*x*	*s*	*V*	*x*	*s*	*V*			
Maximal deviation to the left	−0.93	0.95	−0.84	−1.32	1.16	−1.24	−3.91	<0.001	0.41
Maximal deviation to the right	0.92	1.08	0.87	1.26	1.41	1.27	−4.57	<0.001	0.48
Maximal posterior deviation	1.13	3.50	1.33	1.51	3.42	1.67	−3.46	0.001	0.36
Maximal forward deviation	3.89	3.46	4.41	4.47	3.60	4.85	−3.96	<0.001	0.42
Mean deviation X	0.01	0.83	0.02	0.02	0.87	0.07	−0.26	0.792	0.03
Mean deviation Y	2.67	3.38	2.91	2.76	3.35	3.09	−1.45	0.148	0.15
Mean velocity X	0.85	0.59	0.76	1.45	1.64	1.11	−7.20	<0.001	0.76
Mean velocity Y	1.03	0.71	0.87	1.62	2.38	1.12	−6.56	<0.001	0.69
Path length	32.38	12.84	30.53	56.57	31.78	49.43	−7.41	<0.001	0.78
Surface area	8.12	5.61	7.12	11.86	25.91	6.02	−0.57	0.569	0.06

*x*—mean; *s*—standard deviation; *V*—coefficient of variation; *Z*—Mann-Whitney’s test statistics; *p—p*-value; *r*—effect size.

**Table 3 medicina-56-00505-t003:** Results of statistical analysis of Romberg’s test (eyes open and closed) in the control group (CG).

Romberg’s Test	Eyes Open			Eyes Closed			*Z*	*p*	*r*
	*x*	*s*	*V*	*x*	*s*	*V*			
Maximal deviation to the left	−1.56	2.05	−1.10	−1.75	2.33	−1.34	−1.87	0.062	0.22
Maximal deviation to the right	1.00	1.20	0.78	1.39	1.75	1.14	−3.21	0.001	0.37
Maximal posterior deviation	−0.45	4.48	−0.08	−0.60	4.41	−0.22	−1.04	0.300	0.12
Maximal forward deviation	2.66	4.86	2.71	2.96	4.88	3.36	−1.64	0.101	0.19
Mean deviation X	−0.14	0.65	−0.11	−0.10	0.77	−0.15	−0.68	0.496	0.08
Mean deviation Y	1.01	4.28	1.27	1.22	4.40	1.25	−0.45	0.653	0.05
Mean velocity X	0.71	0.27	0.67	1.13	0.68	0.96	−6.33	<0.001	0.74
Mean velocity Y	0.76	0.32	0.71	1.31	0.76	1.13	−6.48	<0.001	0.75
Path length	29.10	13.13	25.42	48.66	28.23	41.69	−6.10	<0.001	0.71
Surface area	7.48	10.40	4.08	6.14	4.58	4.91	−0.33	0.744	0.04

*x*—mean; *s*—standard deviation; *V*—coefficient of variation; *Z*—Mann–Whitney’s test statistics; *p—p*-value; *r*—effect size.

**Table 4 medicina-56-00505-t004:** Comparison of the Romberg’s test (eyes open) results in the CG group and the control group (CG).

Romberg’s Test-Eyes Open	MAS Group			CG Group			*Z*	*p*	*r*
	*x*	*s*	*V*	*x*	*s*	*V*			
Maximal deviation to the left	−0.93	0.95	−0.84	−1.56	2.05	−1.10	−2.25	0.025	0.18
Maximal deviation to the right	0.92	1.08	0.87	1.00	1.20	0.78	−0.12	0.908	0.01
Maximal posterior deviation	1.13	3.50	1.33	−0.45	4.48	−0.08	−3.00	0.003	0.23
Maximal forward deviation	3.89	3.46	4.41	2.66	4.86	2.71	−2.18	0.029	0.17
Mean deviation X	0.01	0.83	0.02	−0.14	0.65	−0.11	−1.14	0.256	0.09
Mean deviation Y	2.67	3.38	2.91	1.01	4.28	1.27	−2.71	0.007	0.21
Mean velocity X	0.85	0.59	0.76	0.71	0.27	0.67	−1.37	0.170	0.11
Mean velocity Y	1.03	0.71	0.87	0.76	0.32	0.71	−2.78	0.005	0.22
Path length	32.38	12.84	30.53	29,10	13.13	25,42	−2.11	0.035	0.16
Surface area	8.12	5.61	7.12	7.48	10.40	4.08	−2.60	0.009	0.20

MAS—mastectomy; CG—control group; *x*—mean; *s*—standard deviation; *V*—coefficient of variation; *Z*—Mann–Whitney’s test statistics; *p*—*p*-value; *r*—effect size.

**Table 5 medicina-56-00505-t005:** Comparison of the Romberg’s test (eyes closed) results in the MAS group and the control group (CG).

Romberg’s Test-Eyes Closed	MAS Group			CG Group			*Z*	*p*	*r*
	*x*	*s*	*V*	*x*	*s*	*V*			
Maximal deviation to the left	−1.32	1.16	−1.24	−1.75	2.33	−1.34	−0.64	0.521	0.05
Maximal deviation to the right	1.26	1.41	1.27	1.39	1.75	1.14	−0.25	0.804	0.02
Maximal posterior deviation	1.57	3.42	1.51	−0.60	4.41	−0.22	−2.72	0.007	0.21
Maximal forward deviation	4.47	3.60	4.85	2.96	4.88	3.36	−2.42	0.015	0.19
Mean deviation X	0.02	0.87	0.07	−0.10	0.77	−0.15	−1.01	0.310	0.08
Mean deviation Y	2.76	3.35	3.09	1.22	4.40	1.25	−2.61	0.009	0.20
Mean velocity X	1.45	1.64	1.11	1.13	0.68	0.96	−1.25	0.213	0.10
Mean velocity Y	1.62	2.38	1.12	1.31	0.76	1.13	−0.34	0.736	0.03
Path length	56.57	31.78	49.43	48.66	28.23	41.69	−2.00	0.046	0.16
Surface area	11.86	25.91	6.02	6.14	4.58	4.91	−1.69	0.091	0.13

MAS—mastectomy; CG—control group; *x*—mean; *s*—standard deviation; *V*—coefficient of variation; *Z*—Mann–Whitney’s test statistics; *p—p*-value; *r*—effect size.

**Table 6 medicina-56-00505-t006:** Statistical analysis of Romberg’s test results for patients after mastectomy (opened and closed eyes) in terms of the operated side.

Romberg’s Test	Operated Breast Right (*n* = 45)			Operated Breast Left (*n* = 45)			*Z*	*p*	*r*
	*x*	*s*	*V*	*x*	*s*	*V*			
Maximal deviation to the left (open eyes)	−0.97	1.00	−1.03	−0.89	0.91	−1.02	−0.52	0.603	0.05
Maximal deviation to the right (open eyes)	0.92	1.33	1.44	0.91	0.78	0.85	−0.40	0.609	0.04
Maximal deviation to the left (closed eyes)	−1.29	1.32	−1.02	−1.35	1.00	−0.74	−0.45	0.654	0.05
Maximal deviation to the right (closed eyes)	1.34	1.75	1.31	1.19	0.97	0.82	−0.13	0.900	0.01

*x*—mean; *s*—standard deviation; *V*—coefficient of variation; *Z*—Mann–Whitney’s test statistics; *p*—*p*-value; *r*—effect size.

**Table 7 medicina-56-00505-t007:** Results of Romberg’s tests performed with opened and closed eyes in women from the MAS group.

Romberg’s Test	BMI < 25 (*n* = 20)			BMI > 25 (*n* = 70)			*Z*	*p*	*r*
	*x*	*s*	*V*	*x*	*s*	*V*			
Maximal deviation to the left (open eyes)	−0.98	1.09	−1.11	−0.91	0.92	−1.00	−0.40	0.691	0.04
Maximal deviation to the right (open eyes)	0.98	0.98	1.00	0.90	1.11	1.24	−0.82	0.415	0.09
Maximal deviation to the left (closed eyes)	−1.57	1.31	−0.84	−1.24	1.12	−0.90	−1.00	0.317	0.11
Maximal deviation to the right (closed eyes)	1.24	1.37	1.11	1.27	1.43	1.13	−0.58	0.560	0.06

*x*—mean; *s*—standard deviation; *V*—coefficient of variation; *Z*—Mann–Whitney’s test statistics; *p—p*-value; *r*—effect size; BMI—Body Max Index.

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
