# Peer review of "Assessment of Postural Balance in Women Treated for Breast Cancer"

_medicina, 2020, doi:10.3390/medicina56100505_

Round 1

Reviewer 1 Report

This is an interesting original study assessing the postural balance of patients who had mastectomy versus case control patients without breast surgery.

The Authors were able to find a significant difference in the two groups with worse outcome 5-6 years after surgery in mastectomy patients.

The control group matches very well the mastectomy group and the statistical analyst is adequate.    

In total 169 women were enrolled into the study, 74 of them into the control group.

The conclusion of the study was that mastectomy can impair the body balance and this conclusion is supported by the Authors’ data.

Some points should be better discussed in the material and methods as well as in the discussion.

Not clear how was the selection criteria of women with mastectomy. All consecutive mastectomies without reconstruction from February to October 2019? Please specify.

Reconstruction of the breast is the standard of care and it was also in 2019. Please specify why the Patients didn’t have reconstruction (patient’s choice, lack of plastic resources in the Hospital/Country?)

This point would deserve better comments in the discussion.

Interesting would be also a comment into discussion about bilateral mastectomy (contralateral prophylactic) more and more used in USA. Would prophylactic mastectomy achieve better patient balance?

Any data on difference in the balance accordingly to BMI and height in a multivariate analysis?

Author Response

Dear Reviewer #1,

Thank you for reviewing our article titled ‘ Assessment of postural balance in women treated for breast cancer’ We deeply appreciate your opinion as well as constructive comments that contributed to more profound consideration of issues addressed in ourpublication. The comments in your review will guide us in our future work.

            In response to those commentaries we clarified the Materials and Methods section, where we described precisely the process of patient qualification to the procedure patients qualified for our study underwent mastectomy between January and December 2014. In the time of testing, all our research participants were 5 to 6 years afterthis surgical procedure.

We have also improved the discussion section:

  • We explained that IBR (immediate breast reconstruction) is a very rare procedure in Poland, as it only concerns 20-40% of patients. Some reasons why Polish women resign from breast reconstruction are: fear of another surgical intervention, fear of surgical complications and hospitalization, age, fear of disease recurrence, as well as economic issues. Women are also inadequately informed about the course of breast reconstruction surgery and its consequences. Some patients do not decide for this procedure due to their advanced age and the fact that breast reconstruction may be perceived as an act of vanity.
  • We added a comment about bilateral (contralateral prophylactic) mastectomy, since we have observed a rise in the number of bilateral mastectomies performed in recent years, especially in the United States.
  • We have not found any studies evaluating the effect of bilateral (contralateral prophylactic) mastectomy on body posture and balance in the available literature. However, it is reasonable to assume that patients who received this treatment may experience improvement in posture and balance. Moreover, it is important to consider the type of surgical intervention in the axillary fossa.
  • Following your suggestion, we have conducted a multivariate analysis of body balance parameters in relation to BMI. Detailed analysis of results has shown no significant differences between experimental groups in terms of maximum sways to the left and right in tests performed both with opened and closed eyes. It means that, regardless of BMI, patients from the MAS experimental group achieved similar results in terms of maximum sways to the left and right.

Thanks to the comments received, we were able to refine the publication in terms of the content as well as the language. All errors mentioned in the review were corrected in the final version of the publication.

                                                                                                          Kind regards,

                                                                                                          the Authors

Reviewer 2 Report

This is an interesting and understudied area of research that could be important to breast cancer patients.  Nevertheless, I was disappointed that there were very few take-away conclusions except that breast cancer patients in general performed worse than the control patients.  The real question is why?  The authors seem to suggest that it is due to asymmetry because one breast was removed.  But it just as easily could have been due to the chemotherapy, many of which are neuro-toxic, or the long term results on strength and lifestyle caused by the general deconditioning of surgery, chemotherapy, radiation, and the disease itself.  The authors cite studies suggesting that the results might be improved with breast conservation surgery or with reconstruction.  If so, it would be nice to see these groups as controls, rather than just normal people who haven't had cancer.  If differences were shown, it would provide useful results with which to weigh treatment options. Similarly, were the results influenced more by the mastectomy or was it the axillary dissection?  It would be nice to compare the groups with axillary dissection vs SLNB.  Even the question regarding benefit of a breast prosthesis cannot be answered since all patients wore their prosthesis during the study.

I'm not too familiar with the details of the Romberg's test.  However, it seems that the difference between cancer patients and control patients was greater for backward and forward deviation than for left or right deviation.  Is this what we would expect if it were caused by a left-right asymmetry?  It would be nice to analyze women with left and right breast cancer separately to see if this correlated with left or right deviation.  Women with large breasts certainly have more asymmetry that those with small breasts. It would be nice to analyze the results by BMI, as this may be a correlate for breast size.  

Author Response

Dear Reviewer #2,

Thank you for reviewing our article titled ‘ Assessment of postural balance in women treated for breast cancer’ We deeply appreciate your opinion as well as constructive comments that contributed to more profound consideration of issues addressed in our publication. The comments in your review will guide us in our future work.

            In response to those commentaries we clarified the results section.

We fully agree with the reviewer’s opinion that poor results of patients who underwent mastectomy may have been influenced not only by the breast removal procedure itself, but also by chemotherapeutic toxicities, lack of physical exercise or the effects of radiation therapy.

Although the idea of comparing balance parameters in women after mastectomy who either did or did not undergo simultaneous breast reconstruction makes complete sense, it would require us to conduct additional research at this point. Some studies have shown that the procedure of breast amputation coupled with wearing an external breast prosthesis is associated with better posture parameters, which has been emphasized in the discussion section of our article.

The majority of our experimental group (75 patients) underwent ALND procedure, while SLNB was performed only on 15 patients. However, it is fair to say that the analysis of balance parameters in relation to the type of surgical intervention in the axillary fossa might result in some interesting observations. We are currently working on publishing another study, in which we examine the correlation between posture parameters in women after mastectomy and the type of surgical procedure performed in the axillary fossa. This analysis has shown that patients who undergo SLNB have better outcomes of balance tests.

            Following your suggestion, we have conducted a multivariate analysis of body balance parameters in relation to BMI and operated side. Detailed analysis of results has shown no significant differences between experimental groups in terms of maximum sways to the left and right in tests performed both with opened and closed eyes. It means that, regardless of BMI, patients from the MAS experimental group achieved similar results in terms of maximum sways to the left and right.

Thanks to the comments received, we were able to refine the publication in terms of the content as well as the language. All errors mentioned in the review were corrected in the final version of the publication.

                                                                                                          Kind regards,

                                                                                                          the Authors

Round 2

Reviewer 2 Report

I still have a little trouble understanding the significance of the article, but I think the authors have done a good job fixing up everything that it is possible for them to do.